# X-FuseRLSTM: A Cross-Domain Explainable Intrusion Detection Framework in IoT Using the Attention-Guided Dual-Path Feature Fusion and Residual LSTM

**DOI:** 10.3390/s25123693

**Published:** 2025-06-12

**Authors:** Adel Alabbadi, Fuad Bajaber

**Affiliations:** Faculty of Computing and Information Technology, King Abdulaziz University, Jeddah 21589, Saudi Arabia; fbajaber@kau.edu.sa

**Keywords:** neural networks, transformer, encoder, principal component analysis, attack, denial of service, internet of things, internet of things security, intrusion detection, explainable AI

## Abstract

Due to domain variability and developing attack tactics, intrusion detection in heterogeneous and dynamic IoT systems is still a crucial challenge. For cross-domain intrusion detection, this paper proposes a novel algorithm, X-FuseRLSTM, a dual-path feature fusion framework that is attention guided and coupled with a residual LSTM architecture. The proposed algorithm is the combination of four major steps: first, feature extraction using deep encoder and sparse transformer; second, feature fusion of the extracted features and reducing the fused features; third, the classification model; and last, explainable artificial intelligence (XAI). The classification model used is a deep neural network and residual long short-term memory (RLSTM). The model effectively incorporates both spatial and temporal correlations in network traffic data, which improves its detection capability. The model predictions are explained using the XAI techniques. Extensive experiments on datasets including TON_IoT Network, NSL-KDD, and CICIoMT 2024 with both 19-class and 6-class variations show that X-FuseRLSTM achieves the highest accuracy of 99.40% on network, 99.72% on NSL-KDD, and 97.66% for 19-class and 98.05% for 6-class on CICIoMT 2024 datasets. The suggested method is appropriate for practical IoT security applications since it provides strong domain generalization and explainability while preserving computational efficiency.

## 1. Introduction

The Internet of Things (IoT) has garnered substantial momentum in recent years and has become integral to our daily existence [1]. The expansion of IoT devices has resulted in numerous security vulnerabilities, rendering these devices susceptible to various malicious attacks [2]. It is very important to find and fix potential security breaches so that IoT networks are safe and private.

Identifying weaknesses inside ecosystems linked to the IoT would help one to forecast future security risks and carry out proactive actions to prevent undesired incursions. Conventional intrusion detection systems (IDS) are frequently implemented. However, they frequently struggle with interpretability, i.e., the ability to comprehend why the model predicted specific feature values and misclassification attacks. The opaque nature of IDS outputs makes it difficult for cybersecurity experts to grasp the reasons behind warnings, determine the importance of risks, and respond properly. The absence of automated systems capable of generating highly precise results for the detection of assaults is a significant barrier to the prompt identification, diagnosis, and transparency of vulnerabilities that are unique to the IoT [3].

Despite the fact that deep learning (DL) and machine learning (ML) have shown a lot of promise in enhancing cybersecurity for Internet of Things environments, there are several major barriers to their widespread application. IoT devices sometimes struggle to run deep learning models owing to resource limitations, including memory, processing power, and battery life, as well as the computational complexity and substantial hardware support these models demand [4]. The diversity of IoT devices and communication protocols complicates the collection and normalization of training data, which is essential for efficient model learning. The deployment of supervised DL is further hampered in IoT scenarios by the fact that large, labeled datasets are frequently unavailable, unbalanced, or domain specific [5]. Furthermore, because deep learning models cannot be deciphered, they are frequently referred to as “black boxes”. This constraint reduces their usefulness in real-time threat assessments and decision making [6]. This opacity is a big challenge in cybersecurity because human-in-the-loop systems need explainability and trustworthiness [7]. Ultimately, it is constantly difficult for academics to discover methods to create models robust enough to resist evolving attack patterns and hostile inputs since IoT systems are continually developing and exposed to new sorts of threats [8].

This paper proposes a novel hybrid approach that uses an attention-based feature fusion technique along with residual recurrent neural networks (RNNs) to detect different types of attacks in various scenarios. The major contributions of this paper are given below.

Multi-Stage Deep Learning Pipeline for IoT IDS: A novel architecture that combines several DL components, such as deep encoder, sparse transformer, and long short-term memory (LSTM) with skip connections, is proposed for efficient intrusion detection across attack classes in the IoT domain.Balancing the dataset: A preprocessing strategy guarantees that the dataset is balanced across classes. The dataset is balanced using the over-sampling or under-sampling techniques.Dual-Path Feature Extraction using Deep Encoder and Sparse Transformer: The high-dimensional features are learned from the input using a deep encoder for dense feature learning and a Sparse Transformer for collecting long-range relationships and attention-based features.Feature Fusion and Reduction: The features extracted using the dual path are fused and then further reduced using principal component analysis (PCA). This speeds up the computations and selects only the most important information from the huge data.Sequential Modeling Using LSTM and Skip Connections for Attack Classification: An LSTM network integrated with skip connections categorizes different attack types. Skip connections, also known as residual connections, are useful for improving data flow through deeper networks and reducing the impact of vanishing gradients.Use of Explainable Artificial Intelligence (XAI): The prediction made by the proposed model is explained using XAI methods, such as SHapley Additive exPlanations (SHAP) and Local Interpretable Model-agnostic Explanations (LIME), to build trust in the detection system and provide the reason behind the explanation.Generalized Intrusion Detection System Framework Across Various IoT Security Domains: The efficiency of the proposed model is evaluated across heterogeneous IoT datasets, such as NSL-KDD, a general IDS dataset, CIC-IMoT2024, a medical IoT device IDS dataset, and TON_IoT, a network-based IoT intrusion dataset, demonstrating its domain-agnostic design. These datasets differ in attack taxonomies and feature spaces; the proposed model is able to learn transferable patterns of malicious behavior.

## 2. Literature Review

Many researchers are interested in developing DL-based approaches for IDS in IoT devices. Cao et al. [9] suggested a method that combines various datasets for simulating complicated IoT situations, which includes both the IoT device dataset and the network dataset. A cyclic encoding is employed for temporal properties using sine and cosine components. Additionally, the data imbalance issue is resolved by using a hybrid over-sampling and under-sampling technique. In order to create binary classifiers, sort data, and differentiate between attack and non-attack data, a variety of ML and DL algorithms are used. The recommended approach achieved the highest accuracy in multi-class classification at an accuracy rate of 98.87%.

Wang et al. [10] present TBCLNN, a lightweight intrusion detection model based on self-knowledge distillation that enhances detection accuracy while reducing model parameters and computing cost. The binary Harris Hawk optimization (BHHO) algorithm is employed to reduce the dimensionality of traffic features. The lightweight neural network models have residual and inverse residual structures. A lightweight convolution, like a tied block convolution, is used to develop the model. To address the issue of sample imbalance and make up for the performance drop induced by lightweight neural networks, a better self-knowledge distillation loss function is introduced. The accuracy of 99.10% is achieved on the TON_IoT network dataset, 99.92% on the Bot_IoT, and 99.71% on the CIC_IDS2017.

Similarly, Logeswari et al. [11] recommended an IDS that consists of data pre-processing, feature selection, and detection as the three essential subsystems. In order to identify the most pertinent features, a feature selection subsystem that is both robust and employs the synergistic dual-layer feature selection algorithm combines advanced model-based techniques, such as ML methods with recursive feature elimination and particle swarm optimization, with statistical methods, including variance thresholding and mutual information. For the purpose of effectively classifying network traffic as either regular or malicious, the classification subsystem makes use of a two-stage classifier that is referred to as LightGBM and XGBoost. Using the TON-IoT dataset and a number of performance measures, the suggested IDS achieves 93.70% accuracy with MLP.

On the other hand, numerous studies, such as Dash et al. [12], have been reported on the dataset NSL-KDD. The authors suggested an optimized LSTM to detect irregularities in network traffic methods with the Salp swarm algorithm (SSA). The simulation findings indicate that SSA-LSTMIDS outperforms all models analyzed in this work, with an accuracy of 97.98% on the NSL-KDD dataset for five-class attack detection.

Moreover, Xue et al. [13] presented a hybrid autoencoder-hybrid ResNet-LSTM. This advanced residual network integrates a novel hybrid autoencoder with an improved LSTM-CNN architecture to augment model detection capabilities and expedite the identification of relevant feature subsets within datasets. In order to minimize data dimensionality and identify an optimal subset, a modified self-encoder that integrates CNN and GRU components is employed to perform initial feature selection within the dataset. The purpose of this assessment is to evaluate a suggested model. The findings of the experiments show that the accuracy rates for intrusion detection are quite high, with 95.7%, 94.9%, and 96.7%, respectively, for the NSL-KDD dataset on five-class attack detection, the UNSW-NB15 dataset, and the CICIDS-2018 dataset.

Same as above, Raghunath et al. [14] uses ML and feature selection techniques for developing IDS. For training, the NSL-KDD dataset is used. Categorical features are turned into numbers, and PCA is used to reduce the number of dimensions. Performance tests are conducted on a number of classifiers, including support vector machines (SVMs), random forest, and linear regression. Out of all the models, the SVM that was optimized with particle swarm optimization (PSO) had the best accuracy, recall, and specificity at 98.5% on five-class attack detection.

Imrana et al. [15] also presented a CNN-GRU-FF that addresses the class imbalance problem in IDS datasets using a modified focal loss function rather than the standard cross-entropy method. Based on the outcomes of the research, it is clear that the CNN-GRU-FF technique achieves a detection rate (recall) of 99.68% on the NSL-KDD dataset for five-class attack detection.

Similar to above, Thakkar et al. [16] applied a fusion-based methodology that integrates autoencoders (AEs) and principal component analysis (PCA) methodologies to reduce features in IDS. The suggested method tries to find both straight-line and curved connections between traits while making the input data less complicated. The accuracy achieved on the NSL-KDD dataset is 82.22%.

Medical IoT devices are highly prone to attacks. However, due to the lack of publicly available datasets, very few studies have implemented the DL-based approach for IDS in medical IoT. Some researchers, such as Akar et al. [17], suggest an approach that utilizes an improved version of the LSTM deep learning algorithm to detect a variety of intrusion threats directed towards Internet of Medical Things (IoMT) devices. The CICIoMT2024 dataset is employed to assess and contrast the suggested algorithm with alternative techniques. According to the results, the recommended method attained a 98% accuracy for 19 classes.

Similar to the above research, Torre et al. [18] introduced a Federated Learning IDS that uses a one-dimensional convolutional neural network (CNN) to detect intrusions in IoT networks effectively and precisely. The model achieves the highest accuracy of 98.20% on the TON_IoT dataset and 97.31% on 87.72% on the CIC IoMT 2024. Alternatively, Bo et al. [19] used a metric-based meta-learning technique. By creating preliminary designs for every sample category, this approach improves the model’s ability to manage situations involving many classes along with the use of the adaptive feature fusion method. The accuracy obtained was 97.78% in multi-class tasks.

Many researchers use XAI for good interpretations of the predictions made by the IDS systems. Mahmoud et al. [20] provide the XI2S-IDS framework, an intelligent 2-stage intrusion detection system that can be interpreted. The XI2S-IDS architecture integrates a dual-stage methodology with SHAP-based elucidations, enhancing both detection and interpretability for infrequent attacks. XI2S-IDS improves transparency in decision making by utilizing SHAP values, which enable security analysts to acquire a clear understanding of the model’s rationale and the significance of features. Tests were carried out on the UNSW-NB15 and CICIDS2017 datasets, which show considerable gains in detection performance by attaining the accuracy of 97% and 99.8% for the multi-class problem.

Similar to the above, Mittal et al. [21] present DLEX-IMD, a deep learning-based ensemble method for detecting IoT malware attacks that has been trained and evaluated on standard datasets. The dependability of the suggested model training is justified by using the benchmark XAI method, i.e., LIME, to explain the performance of the suggested scheme. The suggested method obtains an F1-score of 0.999 and an accuracy of 99.96%. Alabbadi and Bajaber [22] also used the XAI method along with the CNN, DNN, and TabNet models. The recommended methods achieve an accuracy of 99.24% on the TON_IoT network dataset. Alternative to the above studies, researchers like Deng et al. [23,24] have discussed the concerns of security and challenges for the AI agents.

From the literature review, it is identified that the researchers are mainly focusing on using the CNN and LSTM models for developing IDS in IoT. Some methods use feature fusion techniques to merge different types of features to form a common set, and then the common set is applied to the classification algorithm. Optimization techniques like PSO, BHHO, etc., are also used to optimize the features for better performance. As most of the IDS datasets are highly imbalanced, various balancing techniques like over-sampling or under-sampling are used to enhance the training. Moreover, the IDS dataset is of high dimension; hence, the feature reduction technique is used to reduce the features, which helps the model train faster. Still, the challenge of explainability in IDS systems persists, which makes the IDS system more transparent and also provides the reasons for the prediction of the attack made. Despite huge research in the IDS, the research is still lacking in a generalized model that can be adopted in various domains like medical, agricultural, general networks, IoT networks, etc. Hence, this paper proposes a novel cross-domain IDS that can be adapted to all the different domains and provide high performance. Along with this, the proposed method uses attention-based feature fusion techniques to extract the more prominent features and then apply a feature reduction technique, i.e., PCA, to reduce the features. Then, the reduced features are trained using the residual LSTM. Finally, the LIME and SHAP techniques are used to explain the predictions.

## 3. Methodology

The proposed architecture, as given in Figure 1, demonstrates a structured and comprehensive pipeline for a cross-domain network intrusion detection system (NIDS) that is based on explainable artificial intelligence (XAI) and deep learning. Algorithm 1 presents the mathematical formulation of the proposed method, X-FuseRLSTM IDS. There are three main steps to the process, and they are as follows: data preprocessing, feature extraction and fusion, explaining deep learning modeling using reduced features. Each step is necessary to make sure that the data is correct, that features are represented in a useful way, that classification works well, and that decisions are made clearly.

In the initial phase, data pre-processing and cleaning, the pipeline commences with the acquisition of the raw dataset. The quantity of distinct classes in the target variable is determined to assess the diversity and intricacy of the data. The classifications are subsequently assigned to a predetermined set of categories to standardize the classifying process. Due to the frequent class imbalance in network intrusion datasets, data are sampled utilizing suitable balancing strategies, such as oversampling or under-sampling. Label encoding is then used to transform categorical class labels into a numerical format, facilitating compliance with machine learning models. Subsequent to preprocessing, the dataset is partitioned with the 4:1 ratio for the train and test sets.

Feature extraction is the main emphasis of the second phase, which uses a dual-pathway approach to collect various input data forms. The first path uses a deep encoder, which probably shrinks the input features into a hidden, lower-dimensional space while keeping the important data and getting rid of the noise. The second approach makes use of a sparse transformer, an attention-based system that can recognize contextual links and long-range dependencies in data. These two ways work together to get features that are complementary, which makes sure that the feature set is rich and complete.

The third phase, explained deep learning modeling with reduced features, starts with feature fusion, which combines results from the deep encoder and sparse transformer. Principal component analysis (PCA) is implemented to mitigate the dimensionality of this fused feature set, which may be high-dimensional. PCA improves model efficiency and generalization by keeping the most informative elements and eliminating the less pertinent ones. After the features have been reduced, they are sent to an LSTM network that has skip connections. This network is intended to identify sequential patterns and temporal dependencies in the data that are collected from network traffic. In order to train deeper networks effectively, skip links assist in mitigating the vanishing gradient problem.

Following training, the model’s performance is evaluated using classification metrics like ROC curves, precision, recall, and F1-score to determine how well it can detect different kinds of network intrusions. To improve trust and interpretability, the system incorporates XAI approaches such as LIME and SHapley Additive exPlanations (SHAP). These tools offer cybersecurity professionals the ability to more effectively comprehend and verify the system’s decisions by providing them with an understanding of which features influenced the model’s predictions.
**Algorithm 1:** X-FuseRLSTM**Input:** Raw dataset D={X,Y}, where X∈Rⁿˣᵈ are the features and Y∈Zⁿ are class labels.**Output:** Trained model M and Explanation maps for LIME ℇLIME  and SHAP ℇSHAP**Step 1:** Data Pre-processing and Cleaning **Step 1.1.** Map original labels Y to c classes by using the mapping function as Y′=f(Y), where f: C → C′, |C′|=c  **Step 1.2.** Balance the dataset using sampling techniques (e.g., oversampling or undersampling).  **Step 1.3.** Encode class labels numerically: Y″ ∈ {0, 1, ..., c}   **Step 1.4.** Split the dataset into training and testing sets: DTrain={XTrain,YTrain} and DTest={XTest,YTest}**Step 2:** Feature ExtractionApply two feature extractors:   **Step 2.1.** Let ∅DE: Rd→ Rm, be the encoding mapping trained by the Deep Encoder F1=∅DE(XTrain), F1 ϵ Rn×m where n is the number of features and m is a number of samples.  **Step 2.2.** Let ∅ST: Rd→ Rp, be the Sparse transformer feature extractor F2=∅ST(XTrain), F2 ϵ Rn×p where n is the number of features and p is number of samples. **Step 3:** Feature Fusion and Reduction  **Step 3.1.** Concatenate the Features extracted using F=F1+F2, F∈ Rn×(m+p)   **Step 3.2.** Features are reduced using Freduced=PCAF, Freduced∈ Rn×q, where q is the number of reduced dimensions.**Step 4**: Model Training and Classification  **Step 4.1**. Let φRLSTM: Rq→ Rc be the residual LSTM classifier.  **Step 4.2**. Train the classifier on reduced features Y^Train=φRLSTM(Freduced)  **Step 4.3**. To evaluate the loss, the categorical cross-entropy loss is used as Loss=−∑i=1n∑j=1cyij log⁡(y^ij) **Step 4.4**. Optimize the φRLSTM Model using gradient descent to minimize the Loss**Step 5:** Model Evaluation  **Step 5.1.** Extract the features Ftest from the XTest using steps 2 and 3.  **Step 5.2.** Predict the attacks using Y^Test=φRLSTM(Ftest)  **Step 5.3.** Compute the performance using benchmarked metrics**Step 6:** Explainability with XAI  **Step 6.1.** For the sample xi∈ XTest: ℇLIMExi=LIME(φRLSTM, xi)  **Step 6.2.** For the sample xi∈ XTest: ℇSHAPxi=SHAP(φRLSTM, xi)


### 3.1. Dataset Description and Pre-Processing

Three different domain open-source datasets are used for experimentation: the TON_IOT network IoT dataset, the NSL-KDD network dataset, and the CICIoTM medical IoT dataset.

#### 3.1.1. TON_IoT Network Dataset

The Cyber Range Lab creates an extensive and authentic dataset, the TON_IoT dataset [25], at the University of New South Wales (UNSW) to support research in IoT security. From a wide variety of smart environment devices, both IoT and non-IoT, it gathers system logs, telemetry data, and network traffic. The dataset comprises more than 211,043 network flow samples that were collected from devices such as smart refrigerators, smart thermostats, and smart lighting. TON_IoT addresses a wide range of attack types, including DoS, DDoS, ransomware, password attacks, backdoors, and injection attacks. The flow records are obtained by processing the network data, which includes 44 features such as source and destination IP addresses, ports, protocols, packet statistics (e.g., the number of bytes and packets), connection durations, and behavioral patterns such as the connection rate per second.

The dataset is preprocessed and cleaned to enhance its quality. Initially, the dataset has 43 features, and 1 target variable is “type”, having 10 different categories. The label encoding is applied to the target variable to convert the categorical variable to a numerical variable as “backdoor”: 0, “ddos”: 1, “dos”: 2, “injection”: 3, “mitm”: 4, “normal”: 5, “password”: 6, “ransomware”: 7, “scanning”: 8, and “xss”: 9. The oversampling technique, synthetic minority oversampling technique (SMOTE), is used to balance the dataset for training the DL methods. Thus, after oversampling, each category of the target variable has 50,000 samples.

#### 3.1.2. NSL-KDD

The NSL-KDD dataset [26] is an improved version of the KDD Cup 1999 dataset, which was initially developed for the purpose of evaluating network-based intrusion detection systems. It fixes multiple problems with the KDD’99 dataset, including duplicate and redundant entries that caused skewed learning and overstated classification accuracy in the past. NSL-KDD makes sure that classifiers are unable to simply memorize numerous examples, which results in a more fair and balanced evaluation. There are 125,973 samples and 43 features in each record in the dataset that describe different elements of network connections. These include the length of the connection, the protocol type, the service, the flag, and other traffic statistics.

The NSL-KDD dataset is pre-processed to map the 21 attack types of the target variable into 5, as shown in Table 1. Thus, after the type mapping, the distribution of samples in each category is shown in Figure 2. As the dataset is highly imbalanced, the SMOTE technique is used to balance the dataset. Initially, there are 77,054 normal samples, 53,385 for dos, 14,077 for probe, 3880 for R2L, and 119 for U2R. Thus, after applying SMOTE, the number of samples in each category is 77,054. The final pre-processing step is to apply the label encoding to convert the target variable Attack into a numerical variable.

#### 3.1.3. CICIoTM 2024

The Canadian Institute for Cybersecurity created the CICIoMT2024 dataset [27] as a complete standard to help researchers learn more about how to make the Internet of Medical Things (IoMT) safer. Data were collected via network taps to monitor traffic across multiple communication protocols often used in healthcare environments, such as Wi-Fi, MQTT, and Bluetooth Low Energy. Aside from capturing device behavior while powered on, idle, active, and interactive, the dataset also tracks device behavior throughout these stages to shed light on normal and unusual behaviors. There are 3,204,537 labeled instances included in the CICIoMT2024 dataset. These instances can be classified as either benign or malicious network traffic. Each instance is identified by 46 attributes taken from raw packet capture (PCAP) files to aid in machine learning-based intrusion detection. Header length, protocol type, duration, and counts of various TCP flags are among these characteristics. Indicators unique to protocols such as HTTP, DNS, and MQTT are also included. The collection comprises communication from 40 IoMT devices, i.e., 25 authentic and 15 simulated functioning via protocols such as Wi-Fi, MQTT, and Bluetooth. It includes 18 unique types of attacks, excluding normal, but the attacks have subtypes, which results in 51 categories of attacks [28].

The dataset is pre-processed to convert the data into two types for 6-class and 19-class problems. The attack type mapping for 6-class, i.e., 5 attacks and 1 normal, and 19-class, i.e., 18 attacks and 1 normal, is given in Table 2 and Table 3, respectively. The ‘*’ means all the protocol types. All the attacks that are unchanged are mentioned in Table 3.

In the 6-class problem, there are 4,779,859 samples of DDoS, 1,805,529 samples of DOS, 262,938 samples of MQTT, 192,732 samples of benign, 103,726 samples of recon, and 16,047 samples of spoofing, as shown in Figure 3. Thus, the first two classes, DDoS and DOS, are undersampled to 902,764 for class balancing and simplicity due to the limited graphical processing units (GPUs). The same is true for the 19-class problem; the dataset is highly imbalanced, as shown in Figure 4. Thus, the “DDoS-UDP”, “DDoS-ICMP”, “DDoS-TCP”, “DDoS-SYN”, “DDoS-UDP”, “Dos-TCP”, ”Dos-SYN”, ”Dos-ICMP”, “Benign” are undersampled to 173,036 samples. Finally, the target variable is encoded to numerical values for training.

### 3.2. Feature Extraction

The major contribution of this paper is feature fusion by extracting features from two different techniques. To achieve it, the deep encoder is applied to extract the first set of features, and the second set of features is retrieved from the sparse transformer.

#### 3.2.1. Deep Encoder

Deep encoders, such as multilayer perceptrons (MLPs) or convolutional neural networks (CNNs), learn hierarchical representations of the input data. In this case, low-level and mid-level features are usually extracted using a deep encoder. For example, in image data, the earliest layers of a CNN could collect basic properties such as edges, corners, and textures [29]. As the layers deepen, the model acquires more abstract and high-level representations, such as entire objects or portions of objects. This kind of hierarchical feature learning is very important for understanding complicated data trends. Figure 5 shows the proposed architecture of the deep encoder used for extracting the first set of features.

#### 3.2.2. Transformer

One of the main benefits of the sparse transformer model is that it can capture contextual linkages and long-range interdependence between features. Deep encoders, on the other hand, are particularly effective at learning hierarchical representations of data. The transformer’s main characteristic is its self-attention mechanism, which allows the model to evaluate each feature’s significance in relation to every other feature, irrespective of where those features are located in the input sequence [30]. This attribute enables transformers to concentrate on various segments of the input according to their contextual significance, which is especially advantageous in situations where comprehending feature relationships is essential.

For instance, when working with structured information like time series, the transformer can accurately represent the connections between features that are not right next to each other [31]. Using this capability, the model is able to grasp the facts on a deeper and larger level. Conversely, deep encoders are generally more effective in modeling local patterns than in modeling long-range dependencies, rendering them less suitable for tasks that require global feature interactions. The proposed sparse transformer architecture for extracting the second set of features is given in Figure 6.

### 3.3. Feature Fusion and Reduction

The extracted set of features is concatenated to form a fused feature. Then these features are reduced by applying the feature reduction technique, PCA [32]. PCA is a popular technique for reducing dimensionality. It works especially well with fused features, which combine numerous feature sets from different sources or modalities. When this happens, the merged feature space usually has a lot of dimensions and may have information that is repeated or linked. To solve this problem, principal component analysis (PCA) sorts the associated features into a new set of independent variables called PCs, which are extracted based on the variation. PCA minimizes feature space dimensionality while keeping the most important information by identifying the top principal components that explain most of the variance of the data. This reduction mitigates the curse of dimensionality and lowers the risk of overfitting, which not only simplifies the computational complexity but also improves the performance of subsequent machine learning models [33]. PCA functions as an essential preprocessing step in feature-level fusion tasks, ensuring that the combined features successfully enhance the learning process.

### 3.4. Deep Learning Modeling

The reduced features are trained using deep neural networks and the customized residual LSTM. The architecture of the DNN model is shown in Figure 7. It consists of 4 dense layers. The major contribution is the ResLSTM architecture, which uses an LSTM model with residual connections. The most optimized model for the ResLSTM is given in Figure 8.

The residual LSTM (ResLSTM) architecture proposed here is intended to improve typical stacked LSTM models by including residual (skip) connections, resulting in more efficient gradient flow and sequential pattern learning. Input sequences with a shape of (1, 91), or one time step with 91 characteristics per instance, are accepted by the network. The initial processing block is an LSTM layer with 128 units that utilizes ReLU activation and is programmed to generate sequences. This layer keeps the sequence dimension while capturing complicated temporal dependencies. The output of this layer, shaped (1, 128), is transmitted to a subsequent LSTM layer with 64 units, utilizing ReLU activation and producing sequences of identical temporal dimensions.

A residual connection is created between the first and second LSTM layers’ outputs to improve information flow. Using a dense layer, the first LSTM output is projected into 64 dimensions to match the second LSTM output. An element-wise addition operation is then used to combine these two tensors. As a result of this skip connection, the model is able to incorporate both low-level and mid-level features, which enhances the robustness of the learning process and prevents vanishing gradients during backpropagation.

By performing a GetItem operation on the second LSTM output, the model is able to collapse the temporal dimension and generate a shape vector (64) after constructing the first residual block. These vectors are then subjected to additional processing by means of a dense layer, which brings their dimensionality down to 32. The outcome of the previous residual addition is subjected to a third LSTM layer with 32 units in parallel. This dense projection and the third LSTM outputs are coupled by a second residual addition to create a deeper feature fusion with recurrent and feedforward transformations.

The last phase of the architecture comprises a dense layer featuring 64 units with ReLU activation to enhance the retrieved feature representation, succeeded by a terminal dense layer with 6 units and softmax activation for multi-class classification. For complicated sequence learning problems, the ResLSTM architecture improves training stability, feature representation, and convergence speed by combining deep recurrent processing with shortcut connections.

### 3.5. Explainable AI

In the context of integrating features from several models or data modalities, dimensionality reduction methods like PCA are frequently utilized to condense the feature space while preserving the majority of the inherent variance. The curse of dimensionality is lessened, and this modification greatly decreases computational complexity. However, it also raises interpretability concerns because the original semantic meaning of characteristics becomes intertwined in the principal components. In order to resolve this issue, XAI techniques, including SHAP and LIME, can be implemented to interpret predictions that are derived from the reduced fused feature space. When explaining specific predictions, the LIME [34] approximation finds the best fit for the model close to the instance. To ascertain which features most substantially impact a specific prediction, a straightforward and interpretable model such as linear regression is trained on data produced by implementing minor modifications to the original input. SHAP [35], on the other hand, is based on joint game theory and calculates Shapley values for each part of a model. This approach takes into account all of the available feature combinations to distribute the contribution of each characteristic to a prediction in an equitable manner. This method ensures that explanations are coherent and backed by theory.

### 3.6. Evaluation Metrics

The standard evaluation metrics used to assess the performance of an intrusion detection system (IDS) include accuracy, recall, precision, and the F1 score. The mathematical expression for accuracy is presented in Equation (1), recall in Equation (2), precision in Equation (3), and the F1 score in Equation (4). In these equations, TP denotes true positives, TN stands for true negatives, FP represents false positives, and FN indicates false negatives.

Accuracy is the proportion of right predictions, encompassing true positives and true negatives, relative to the total number of predictions executed [36].(1)$$Accuracy=\frac{TP+TN}{TP+TN+FP+FN}\;$$

It is the fraction of actual positives that were correctly detected by the model that is measured by the recall mechanism [36]. A high recall indicates proper prediction of the majority of positive cases. It is essential in situations where the cost of missing a positive instance is substantial, such as when diagnosing a sickness or detecting an intrusion.(2)$$Recall=\frac{TP}{TP+FN}$$

Precision is the number of right identifications out of all the ones that were made. High precision results in a reduced number of false positives [36]. When the cost of false positives is considerable, such as in the case of spam detection and fraud detection, it is crucial to avoid them.(3)$$Precision=\frac{TP}{TP+FP}\;$$

The F1 score represents the harmonic mean of precision and recall. It is advantageous in scenarios characterized by an unequal distribution of classes, as it provides a singular measure that satisfies both concerns [36].(4)$$F1\;score=2\times \frac{Precision\times Recall}{Precision+Recall}$$

## 4. Results and Analysis

The experiments were conducted in Python version 3.13.2 on the Kaggle platform, utilizing NVIDIA P100 graphical processing units for computational acceleration. Table 4 compares all the models with respect to different domain datasets. Overall, X-FuseRLSTM regularly does better than X-FuseDNN in terms of all metrics. This shows that adding residual LSTM layers makes it easier to extract and generalize temporal features. With a lower loss (0.0174 vs. 0.0199) and fewer epochs due to early stopping, X-FuseRLSTM achieves a near-perfect accuracy of 99.72% on the NSL-KDD dataset, whereas X-FuseDNN achieves 99.54%. On the network dataset, a similar pattern is observed, where X-FuseRLSTM achieves an accuracy of 99.40% with a training time that is drastically decreased (3.1 min), in comparison to X-FuseDNN, which achieves an accuracy of 94.34% and takes 18 min. Both models perform with well above 97% accuracy on the more complicated CICIoMT 2024 datasets, which include 19 and 6-class labels, respectively. However, X-FuseRLSTM outperforms X-FuseDNN by a little margin across all measures. Due to increasing complexity and lengthier sequence modeling on CICIoMT datasets, X-FuseRLSTM training time increases, but its superior performance justifies the computational expense. In general, X-FuseRLSTM is more reliable and scalable, especially for datasets with complex class structures and time-dependent relationships.

### 4.1. TON_IoT Network

The accuracy and loss graphs for the X-FuseDNN for the network dataset are given in Figure 9.

The training accuracy continuously increases from around 0.89 to 0.934. Still, the validation accuracy fluctuates but constantly remains higher, peaking at 0.945, demonstrating robust generalization without evidence of overfitting, as shown in Figure 9a. In the same vein, the training loss experiences a gradual decrease from above 0.25 to approximately 0.17. In contrast, the validation loss experiences a more abrupt decrease from over 0.30 to approximately 0.13, though with a greater degree of variability, as shown in Figure 9b. Validation loss staying below training loss over time is a sign of good learning and good generalization.

Figure 10 shows the X-FuseRLSTM model’s performance during training and validation on the TON_IoT Network dataset over 186 epochs, as shown through loss and accuracy curves given in Figure 10a,b, respectively. Training and validation accuracy improve rapidly in the first epochs, stabilizing above 99.5% and remaining tightly aligned, showing good generalization with little overfitting. The training and validation loss curves exhibit a steep reduction from approximately 0.7 to below 0.03 early in the training process, then plateauing near zero, indicating consistent and efficient learning. The model’s stability and robustness are demonstrated by the nearly identical trends of the training and validation metrics, which show that the residual LSTM architecture successfully captures temporal patterns in the data.

Figure 11 and Figure 12, respectively, show the areas under the receiver operating characteristic curve (AUC-ROC) for X-FuseDNN and X-FuseRLSTM. All the classes in X-FuseRLSTM have an AUC of 1, while for the X-FuseDNN, all classes except class 3, i.e., injection, and class 6, i.e., password, have an AUC of 1. The diagonal dashed line is the performance of the random classifier with no discriminating capability. 

Figure 13 and Figure 14, respectively, visualize the interpretative insights from two prevalent explainability methods, LIME and SHAP, in the decision-making process of the X-FuseRLSTM framework, specifically for a sample confidently classified as normal. This analysis underscores the significance of various principal components (PCs) in affecting the model’s output, as PCs are given as an input to the residual LSTM.

The LIME plot depicts each feature’s contribution to the model’s final output. It gives a set of conditional thresholds for specific features and assesses their effect on the final result. For example, PC2 > 0.85 and PC4 > 0.81 are displayed in red, which indicates that these values are generally related to attack patterns. However, traits like PC6 ≤ −0.70, PC5 ≤ −0.71, and PC26 ≤ −0.03 appear in green, validating the “normal” classification assumption. The LIME plot’s prevalent green contributions serve as confirmation that, despite the presence of certain features that correspond to attack characteristics, the overall feature combination is more closely aligned with benign behavior, resulting in a prediction probability of 1.00 for the normal class.

However, the same sample is depicted for SHAP explanations as well. The expected output across the dataset is represented by the base value, which begins at a lower probability (near 0). The final model prediction, f(x) = 1.0, indicates a highly confident classification as “normal”. The features PC1 (7.07), PC4 (1.53), and PC2 (3.74) significantly influence this conclusion, as demonstrated by their extensive red bars driving the forecast towards 1.0. These values are indicative of strong positive impacts, which indicates that they are typical of normal behavior in accordance with the patterns that the model has acquired. These characteristics are more likely to be in the “normal” class, as shown by the red shading, and there is little evidence of an attack, due to the lack of noticeable blue segments.

### 4.2. NSL-KDD Dataset

Figure 15 shows the proposed X-FuseDNN model’s training and validation performance on the NSL-KDD dataset, including accuracy and loss over 85 training epochs. The training accuracy is constantly high, reaching 99% during the first 10 epochs and staying there. The validation accuracy is also very close to this trend, suggesting that there is minimal overfitting, as seen in Figure 15a. The related loss curves are shown in Figure 15b. During the initial epochs, both training and validation losses decline dramatically, but then they reach a plateau as training continues until they reach their maximum value. The model seems to be good at generalizing to new data since the validation loss fluctuates slightly but stays low.

The XFuseRLSTM model’s loss curves and training and validation accuracy for the NSL-KDD dataset are shown in Figure 16. With little deviation between the two curves, Figure 16a demonstrates that both training and validation accuracy rise quickly during the first few epochs, achieving more than 99% and then converging. This suggests excellent generalization and stability. Figure 16b shows a significant decrease in both training and validation loss during the initial epochs, which is followed by a consistent and low loss value for the remainder of the training period. The training and validation loss curves have almost similar paths, which suggests that the model is well regularized and does not overfit. The ROC curve for both the proposed models is given in Figure 17, with an AUC of 1 for all classes.

Figure 18 and Figure 19 depict the XFuseRLSTM model’s predictions on the NSL-KDD dataset utilizing LIME and SHAP methodologies, respectively. Figure 18, which was obtained from LIME, demonstrates that the model accurately predicted the sample as “normal” with a probability of 1.00. The main factors that affected this choice are shown as principal components, e.g., PC2, PC4, and PC1. Most of them support the normal class given by the green bars, but PC34 slightly goes against it, as indicated by the red bar. The second image employs SHAP to offer a more detailed explanation, demonstrating how the model output is collectively driven toward the “normal” prediction by individual features, including PC1, PC2, PC4, and PC10. Little blue sections show very little opposing effect. In contrast, large red segments show elements that are pushing the output in the direction of the prediction, which strengthens the model’s confidence.

### 4.3. CICIoMT 2024 Dataset (19-Class)

Figure 20 shows the results of the XFuseDNN model’s training and validation on the CICIoMT 2024 dataset’s 19-class classification job. Figure 20a depicts model accuracy across 54 epochs, revealing that both training and validation accuracies swiftly rise and stabilize between 96.5% and 97%, signifying effective learning and generalization. Figure 20b depicts the loss curves, which indicate that both training and validation losses reduce dramatically during the first epochs before plateauing with small oscillations and remaining low throughout. The model’s robustness and dependability in multi-class intrusion detection are confirmed by the steady alignment of training and validation metrics. This indicates that the model is well optimized and shows no symptoms of overfitting.

Figure 21 shows the XFuseRLSTM model’s training dynamics on the CICIoMT 2024 dataset for the 19-class classification job. Strong generalization is seen by the accuracy graph on the left, which displays a consistent improvement with training and validation accuracy convergent at 97.5%. Despite slight changes in validation accuracy during early epochs, the model remains stable as training advances. The loss graph on the right exhibits a sharp initial decrease in both training and validation loss, thereafter transitioning to a gradual plateau. Around epoch 5, there is a short increase in validation loss that goes away quickly, which suggests that there was only temporary instability, possibly caused by complex class boundaries. The model’s efficacy for challenging intrusion detection tasks is confirmed by its overall excellent performance, convergence stability, and low overfitting.

The ROC curve for both models for the 19-class problem in the CICIoMT 2024 dataset is given in Figure 22. All classes have an AUC of 1, except two classes, Recon-OS_scan and Recon-VulScan, which have an AUC of 0.99.

Figure 23 and Figure 24 depict explainability insights utilizing LIME and SHAP, respectively, for the XFuseRLSTM model on a 19-class intrusion detection task from the CICIoMT 2024 dataset. Based on the LIME explanation, the model predicted the sample as DoS-UDP with a 0.99 probability. This prediction was significantly influenced by features such as PC15 > 0.13, PC26 > 0.05, and PC18 > −0.06, which positively contributed to the prediction (green bars). However, the supporting elements were more important than aspects with negative contributions, such as PC8, PC3, PC49, and PC69 (red bars). The SHAP summary supports PC1, PC3, and PC9, all with severely negative values that contribute heavily to the high model output of 0.99 for DoS-UDP.

### 4.4. CICIoMT 2024 Dataset (6-Class)

The accuracy and loss graphs for the X-FuseDNN are given in Figure 25. Both training and validation accuracy increase swiftly during the initial epochs, stabilizing above 97%, with slight fluctuations in validation accuracy suggesting some sensitivity to class variance; however, no significant overfitting is observed. The loss displays a pronounced early decline in both training and validation loss, which closely converge and stabilize around 0.06, signifying robust generalization and optimal learning parameters.

For the X-FuseRLSTM, the accuracy graph and loss graphs for X-FuseRLSTM for the 6-class problem in the CICIoMT 2024 dataset are given in Figure 26. It reveals that both training and validation accuracy are continuously rising, converging around 98%, although the validation accuracy is lower and has tiny fluctuations, indicating that learning is unstable at times. The loss graph, as shown in Figure 26b, indicates a rapid initial decline, with the training loss attenuating to approximately 0.035. Nevertheless, during training, the validation loss stays larger and more volatile, stabilizing at 0.07, which can suggest some overfitting or sensitivity to particular class distributions. Regardless, the overall high accuracy reveals XFuseRLSTM’s superior classification ability in a reduced-class context.

Figure 27 shows the ROC curve for both models for the 6-class problem, with an AUC of 1 for all 6 classes. Figure 28 and Figure 29 give the LIME and SHAP explanations of the X-FuseRLSTM model, indicating that the top features responsible are PC38, PC5, PC2, PC47, and PC6, as shown in the LIME explanations.

### 4.5. Ablation Study

The ablation study is performed to finalize the proposed architecture given in Figure 1 and the residual LSTM architecture shown in Figure 8. Extensive experiments were performed on all three datasets, i.e., TON_IoT network, KDD, and CICIoMT 2024, with and without PCA. Also, multiple experiments were performed before finalizing the architecture of the proposed model, X-FuseRLSTM, to finalize the best architecture. It was found that X-FuseRLSTM with PCA achieved the highest performance. Table 5 below shows the accuracy of the proposed model with and without PCA. Also, the results are with and without residual connection.

The comparison between the baseline model, which consisted of feature fusion and RLSTM without PCA, and the final model, which consisted of X-Fuse RLSTM with PCA and skip connections, reveals a significant improvement in classification accuracy across all datasets. In particular, TON_IoT saw an improvement in accuracy of 0.17% (from 99.23% to 99.40%), KDD saw an improvement of 0.19% (from 99.53% to 99.72%), CICIoMT (19-class) saw an improvement of 0.59% (from 97.09% to 97.66%), and CICIoMT (6-class) saw an improvement of 0.41% (from 97.64% to 98.05%). These improvements underscore the efficacy of integrating dimensionality reduction with a more expressive architecture that incorporates recurrent learning and skip connections.

The intermediate model, which consists of feature fusion and LSTM with PCA, is compared to the X-Fuse RLSTM, which already incorporates PCA. The additional architectural upgrades, which include the utilization of RLSTM and skip connections, also result in noticeable gains. The accuracy of TON_IoT, KDD, CICIoMT (19-class), and CICIoMT (6-class) increased by 0.07%, 0.05%, 0.39%, and 0.18%, respectively. These findings highlight that, in addition to reducing feature dimensionality, using RLSTM over standard LSTM and including skip connections improves the model’s ability to learn temporal dependencies and hierarchical representations, especially in complex, multi-class intrusion detection tasks.

### 4.6. Comparison of Proposed Model X-FuseRLSTM with Benchmarked Methods

Table 6 shows the comparison of the X-FuseRLSTM with the benchmarked methods for each of the datasets.

Due to the fact that it incorporates attention-guided feature fusion and residual LSTM architecture, the X-FuseRLSTM that has been suggested consistently outperforms other approaches across all datasets. In contrast to traditional models that depend only on convolutional neural networks (CNNs), long short-term memories (LSTMs), or optimization methods, X-FuseRLSTM efficiently captures both spatial and long-term temporal connections. It outperforms hybrid and heuristic-based models with top-tier accuracy on the NSL-KDD and network datasets. In the difficult CICIoMT 2024 dataset, it maintains high performance across both 19- and 6-class scenarios, demonstrating strong generalization. Its capacity to highlight essential features and reduce overfitting enhances its resilience in practical intrusion detection settings.

## 5. Discussion

The X-FuseRLSTM model consistently beats the X-FuseDNN model in all important evaluation measures, according to a performance comparison between the two models across several intrusion detection datasets. This improvement is seen across a range of dataset complexities, including classic benchmarks and more modern, multi-class intrusion detection datasets. X-FuseRLSTM works better than other methods because it has a hybrid architecture that combines residual learning with LSTM-based temporal modeling. This lets it effectively catch both spatial correlations and long-range dependencies in sequential network traffic data. In addition, the attention-guided dual-path feature fusion technique enables the model to concentrate on the aspects that are most important while simultaneously reducing the amount of noise, which improves the model’s ability to generalize across different domains. Even though it takes a little longer to train because of how complicated its architecture is, X-FuseRLSTM does well with early stops and strong convergence. Its steady dominance points to a greater degree of adaptability to diverse IoT traffic patterns, which makes it especially suitable for cross-domain intrusion detection jobs in the real world.

The most powerful use case of the proposed model is in the medical field, where time series data is present. Thus, in the CICIoMT dataset, the X-Fuse RLSTM was able to accurately identify a low-rate DDoS attack that imitated typical traffic patterns by ensuring that packet intervals and payload sizes were consistent. This abnormality went undetected by traditional detectors because it was statistically so similar to normal traffic. Accurate detection was made possible by X-Fuse RLSTM, which used deep temporal features and attention-guided fusion to detect minute variations in connection frequency and timing patterns over lengthy durations. This highlights the model’s capacity to identify sneaky and developing threat behaviors that are difficult to detect using traditional methods.

Still, this research has some limitations, as the methodology for utilizing LIME on PCA-reduced fused features entails handling the principal components as new input features for explanation. LIME changes the reduced feature vector around a forecast and fits a simple model that is easy to understand (like linear regression) to get close to the decision boundary. Although LIME sheds light on the relative importance of the primary components in making a forecast, it should be remembered that it is not always easy to understand the meaning of individual components.

SHAP can also be applied to the fused PCA-reduced features by computing Shapley values, which provide priority to each principal component. SHAP values give a theoretically sound way to measure how important a feature is by taking into account all the different ways that features can contribute. Similar to LIME, the straightforward interpretation of SHAP values concerning principal components might be convoluted. However, despite providing the explanations of the predictions, the original features cannot be tracked as the original features go through a deep encoder, a transformer for feature extraction, and then are fused. The fused features are reduced using the PCA, and the PCs generated are the input to the models like DNN and RLSTM, thus making it impossible to explain the predictions using the original features.

Instead of applying PCA directly to raw input features, the XFuse-RLSTM architecture applies it to fused feature representations produced by deep encoders and a sparse transformer. These fused features are high-level nonlinear embeddings that represent the data’s complicated temporal and spatial interactions. As a result of this abstraction, the principal components derived from PCA are combinations of latent features rather than interpretable original attributes, rendering it impossible to perform a direct post hoc mapping back to raw input features. As a consequence, conventional explainability methods that are dependent on feature-level attribution are subject to constraints in this scenario. As an alternative, the proposed method focuses on analyzing intermediate model components, including attention weights and latent embeddings, to understand the patterns that drive model decisions and tackle interpretability in that way. The proposed method is in line with the state-of-the-art in explainable deep learning; nevertheless, it is understood that there is a need for additional study to create explanations for fused feature-based models in security applications that are both more transparent and have semantic significance.

## 6. Conclusions

Within the scope of this research, X-FuseRLSTM was offered as an innovative and efficient cross-domain intrusion detection framework that was developed for Internet of Things scenarios. There are four main parts to the algorithm: (1) a feature extraction process that utilizes a deep encoder and sparse transformer to obtain rich spatial and temporal representations; (2) dual-path feature fusion with dimensionality reduction to improve the fused features’ discriminative capacity; (3) a hybrid classification model that incorporates DNN and RLSTM to learn intricate temporal patterns in traffic data; and (4) an XAI module to decipher model decisions. This design not only improves the accuracy of detection, but it also keeps things easy to understand, which is very important for real-world use. X-FuseRLSTM achieved 99.40% on the network dataset, 99.72% on NSL-KDD, and 97.66% and 98.05% on CICIoMT 2024 in both 19-class and 6-class settings. These findings provide convincing evidence that the model is superior and can generalize over a wide range of domains. In the future, researchers will focus on making the model easier to understand by adding more features to the XAI module that work with original inputs, along with fused and reduced versions. This will increase transparency and offer deeper insights into decision-making processes, particularly for security analysts working in operational settings.

## Figures and Tables

**Figure 1 sensors-25-03693-f001:**
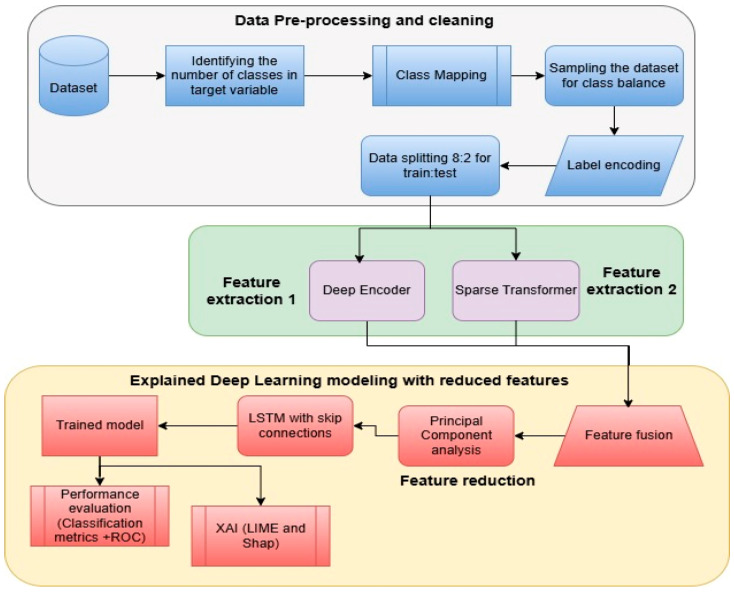
Proposed architecture for the X-FuseRLSTM.

**Figure 2 sensors-25-03693-f002:**
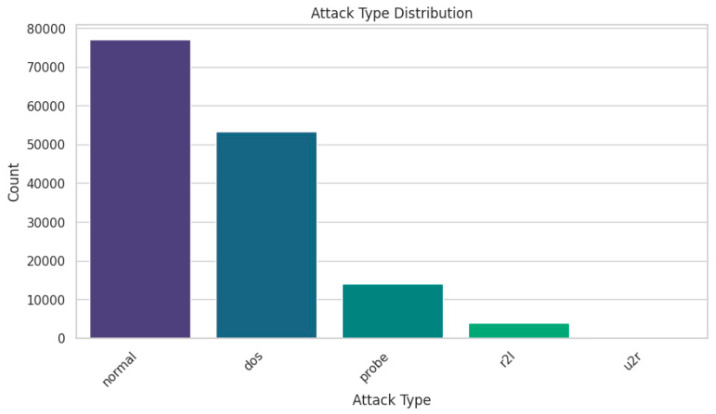
Distribution of samples of NSL-KDD dataset after attack type mapping.

**Figure 3 sensors-25-03693-f003:**
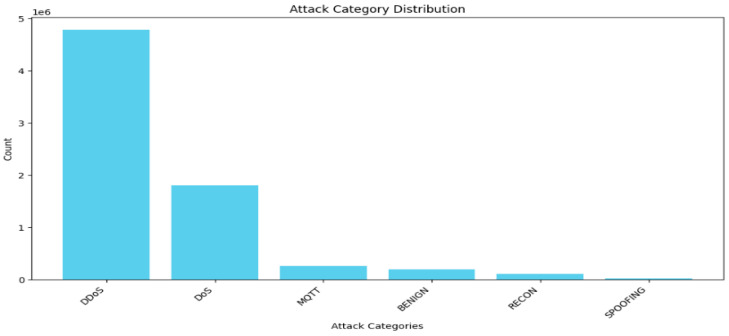
Data distribution after conversion of the CICIoTM 2024 dataset into a 6-class dataset.

**Figure 4 sensors-25-03693-f004:**
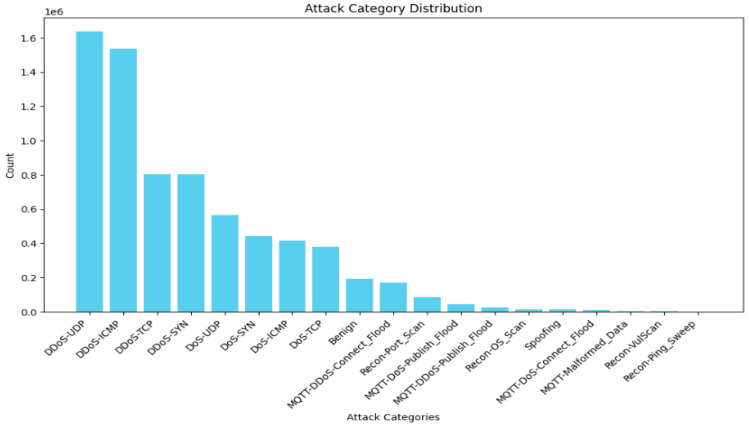
Data distribution after conversion of the CICIoTM 2024 dataset into a 19-class dataset.

**Figure 5 sensors-25-03693-f005:**
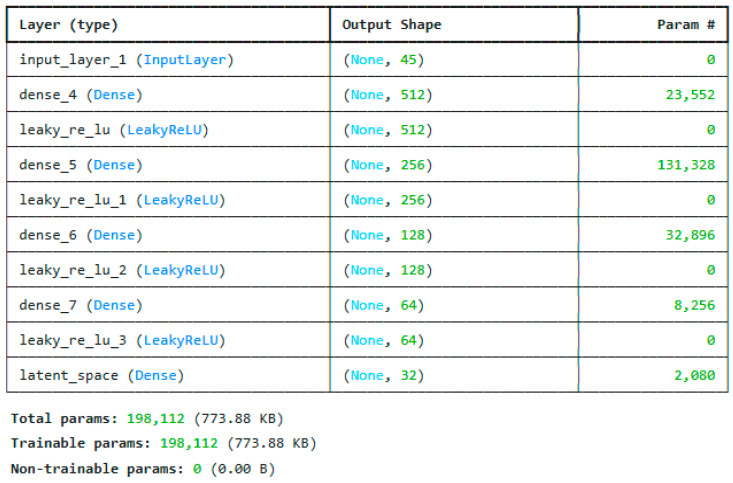
Architecture of the proposed deep encoder as feature extractor for the CICIoTM 2024 dataset.

**Figure 6 sensors-25-03693-f006:**
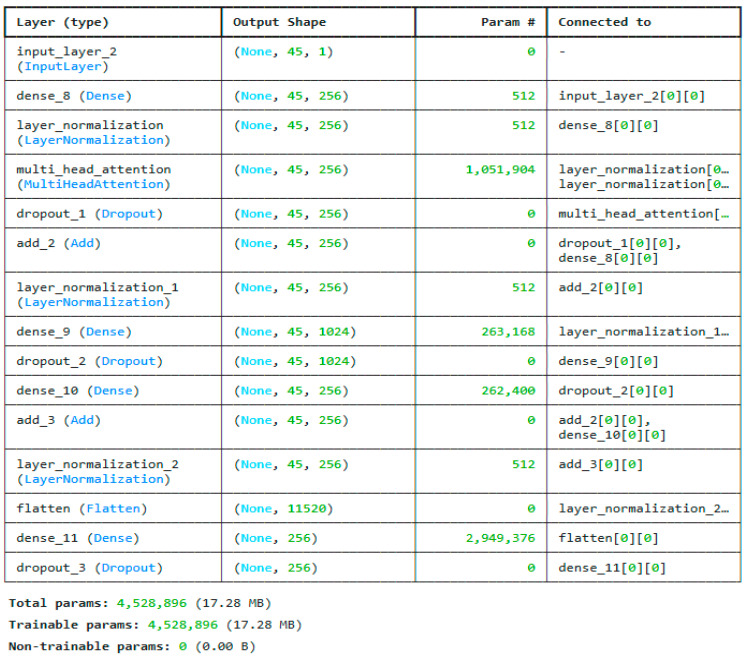
Proposed sparse transformer architecture for extracting features for the CICIoTM 2024 dataset.

**Figure 7 sensors-25-03693-f007:**
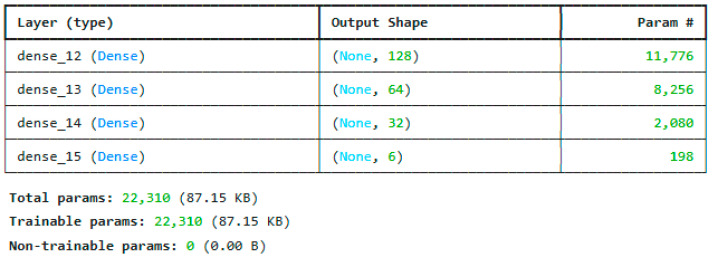
Proposed DNN architecture.

**Figure 8 sensors-25-03693-f008:**
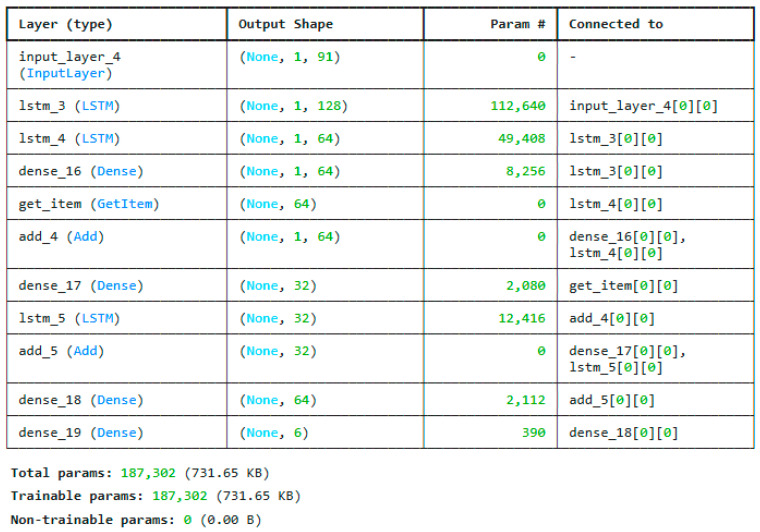
Proposed ResLSTM architecture.

**Figure 9 sensors-25-03693-f009:**
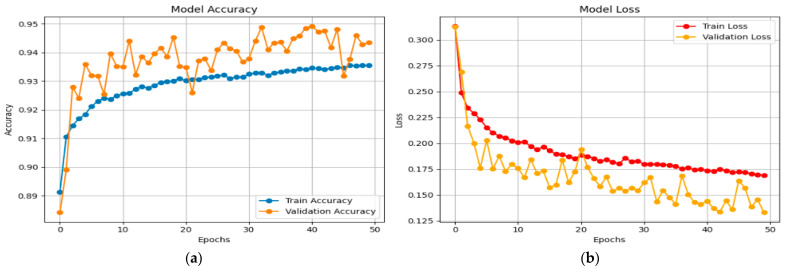
(**a**) Accuracy and (**b**) loss graphs for X-FuseDNN for TON_IoT network dataset.

**Figure 10 sensors-25-03693-f010:**
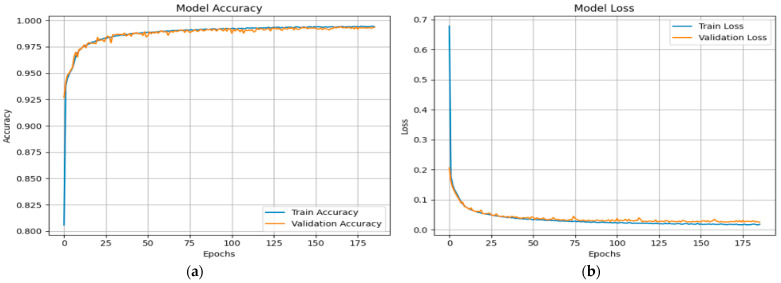
(**a**) Accuracy and (**b**) loss curves for X-FuseRLSTM for TON_IoT dataset.

**Figure 11 sensors-25-03693-f011:**
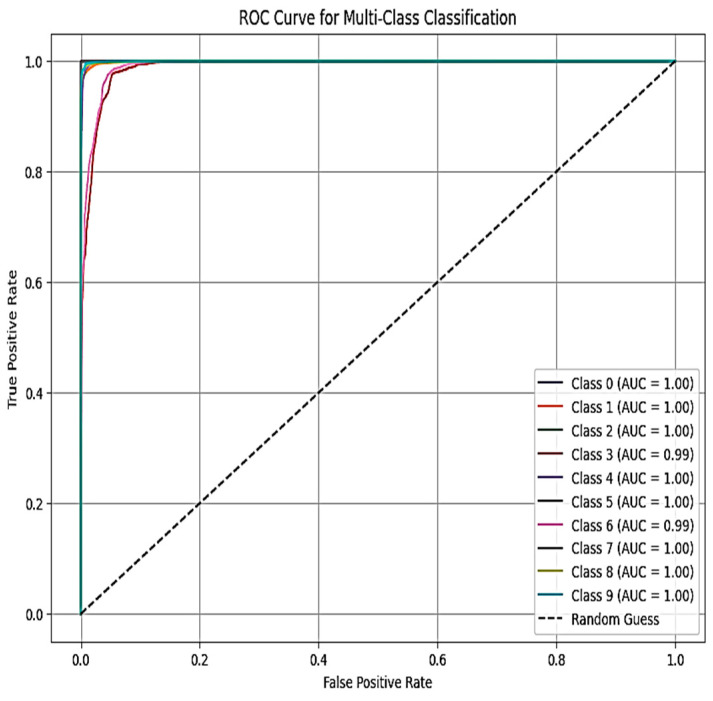
ROC curve for the TON_IoT dataset for X-FuseDNN.

**Figure 12 sensors-25-03693-f012:**
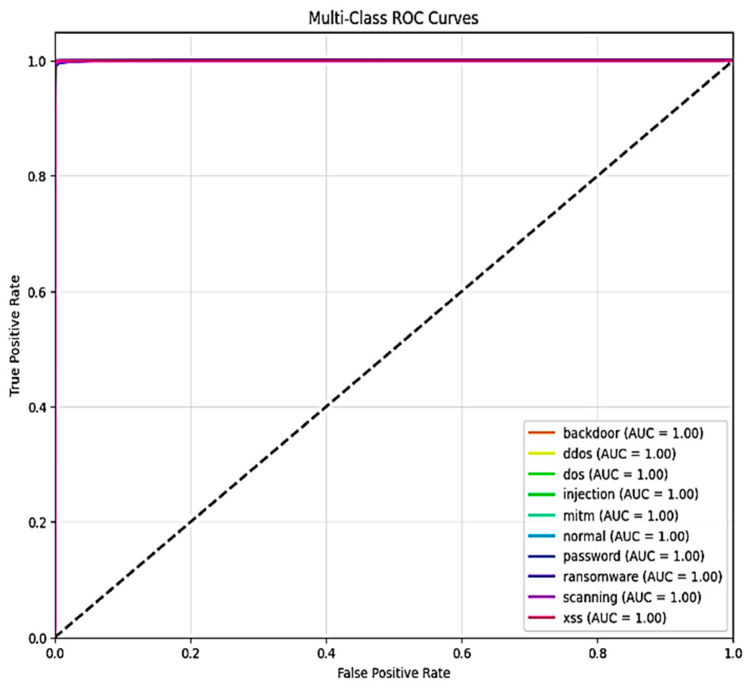
ROC curve for TON_IoT dataset for X-FuseRLSTM.

**Figure 13 sensors-25-03693-f013:**
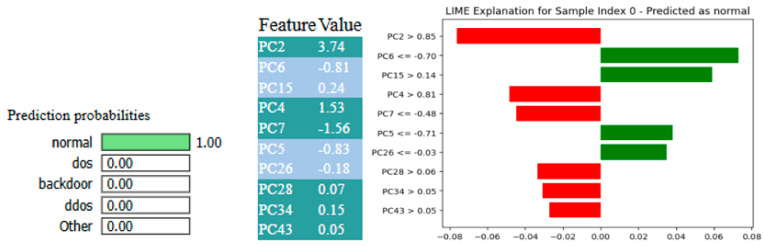
XAI-LIME explanations for X-FuseRLSTM for TON_IoT dataset.

**Figure 14 sensors-25-03693-f014:**
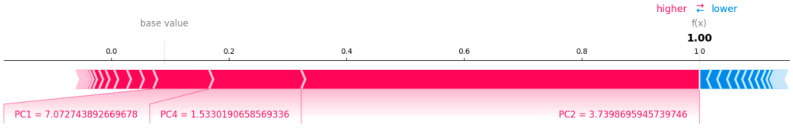
XAI-SHAP explanations for X-FuseRLSTM for TON_IoT dataset.

**Figure 15 sensors-25-03693-f015:**
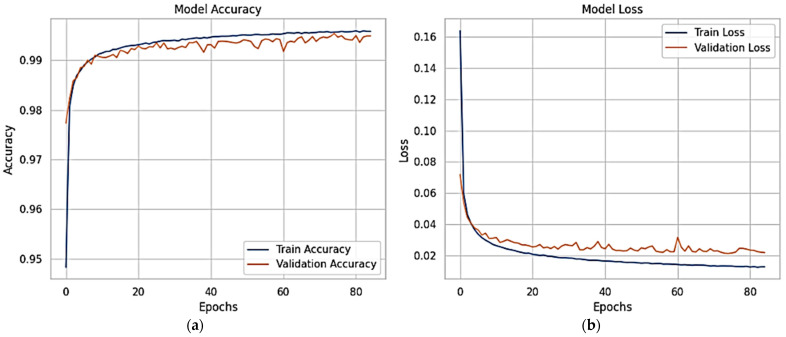
(**a**) Accuracy graphs and (**b**) loss graphs for X-FuseDNN for the NSL-KDD dataset.

**Figure 16 sensors-25-03693-f016:**
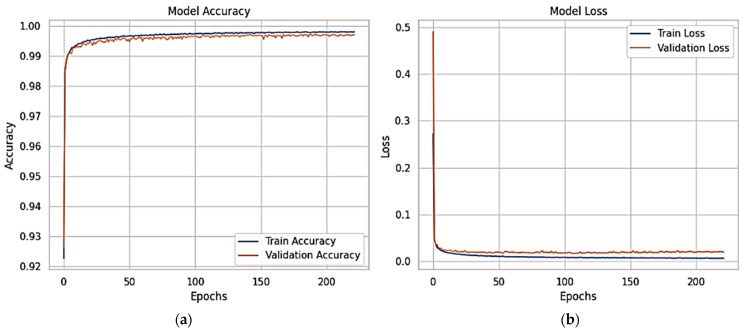
(**a**) Accuracy graphs and (**b**) loss graphs for XFuseRLSTM for the NSL-KDD dataset.

**Figure 17 sensors-25-03693-f017:**
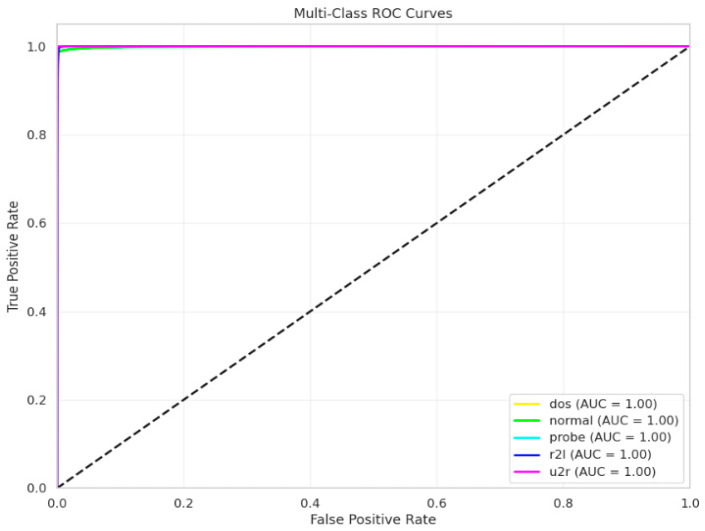
ROC curve for NSL-KDD for both proposed models.

**Figure 18 sensors-25-03693-f018:**
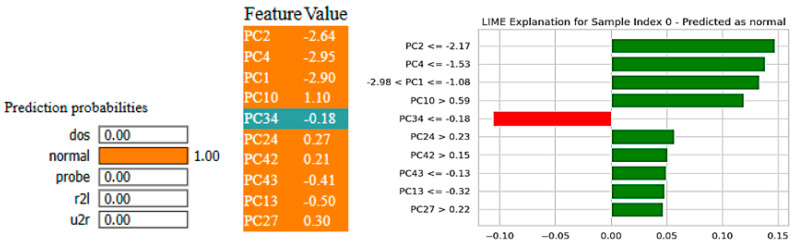
XAI-LIME for XFuseRLSTM for NSL-KDD dataset.

**Figure 19 sensors-25-03693-f019:**
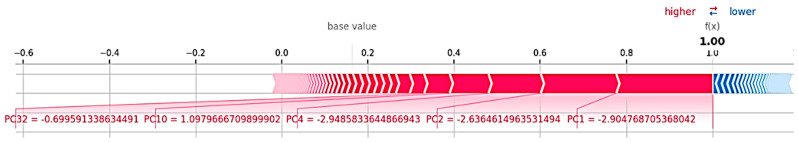
XAI-SHAP explanations for XFuseRLSTM for NSL-KDD dataset.

**Figure 20 sensors-25-03693-f020:**
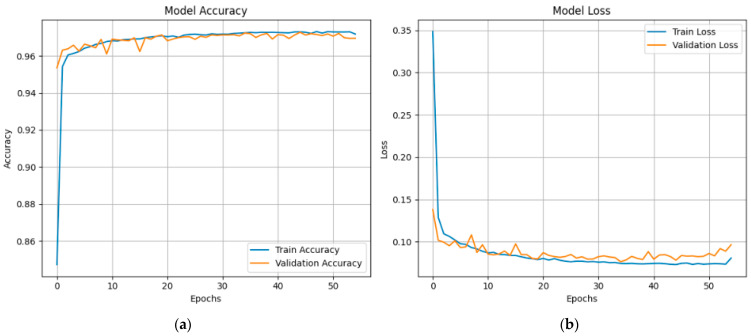
(**a**) Accuracy and (**b**) loss graphs for XFuseDNN for 19-class in CICIoMT 2024.

**Figure 21 sensors-25-03693-f021:**
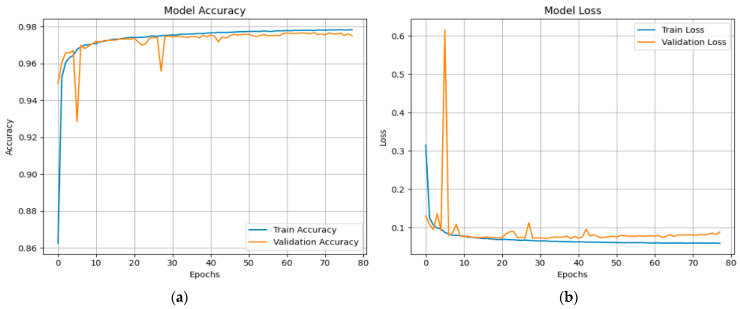
(**a**) Accuracy and (**b**) loss graphs for XFuseRLSTM for 19-class in CICIoMT 2024.

**Figure 22 sensors-25-03693-f022:**
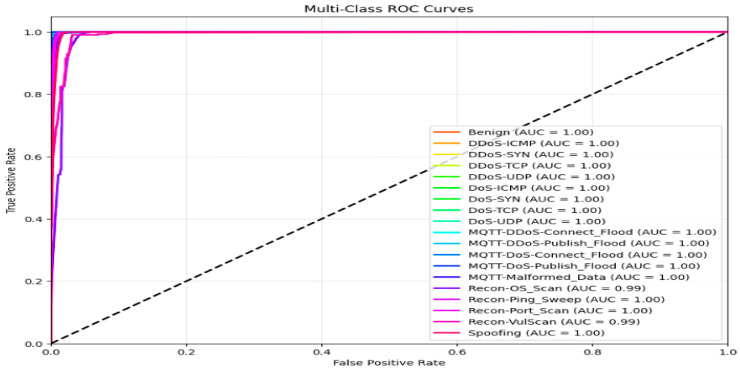
ROC curve for both the proposed models for the 19-class in CICIoMT.

**Figure 23 sensors-25-03693-f023:**
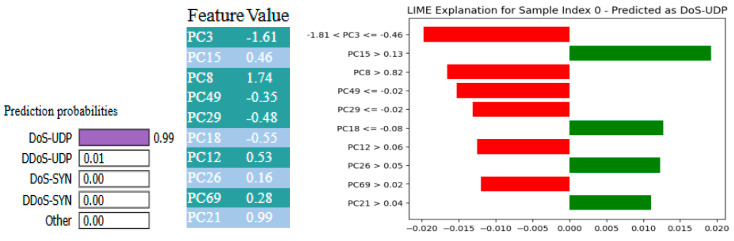
XAI-LIME explanations for XFuseRLSTM for 19-class in CICIoMT 2024.

**Figure 24 sensors-25-03693-f024:**
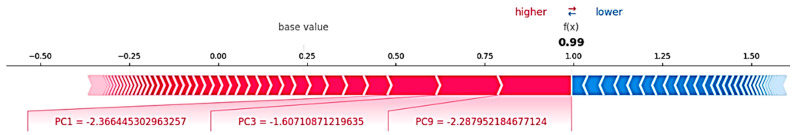
XAI-SHAP explanations for XFuseRLSTM for 19-class in CICIoMT 2024.

**Figure 25 sensors-25-03693-f025:**
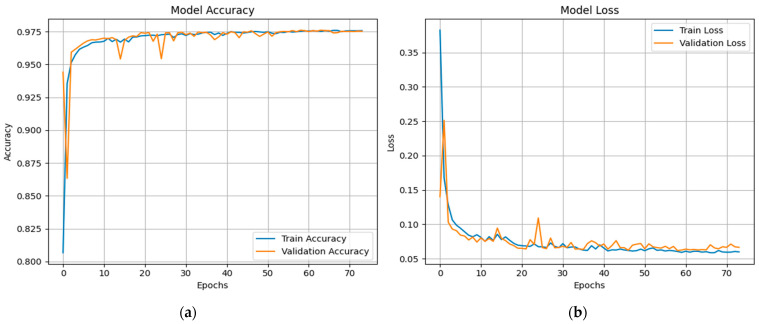
(**a**) Accuracy and (**b**) loss graphs for XFuseDNN for 6-class in CICIoMT 2024.

**Figure 26 sensors-25-03693-f026:**
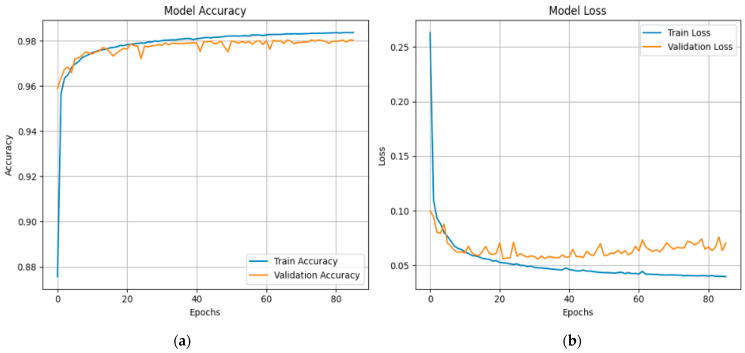
(**a**) Accuracy and (**b**) loss graphs for X-FuseRLSTM for 6-class in CICIoMT 2024.

**Figure 27 sensors-25-03693-f027:**
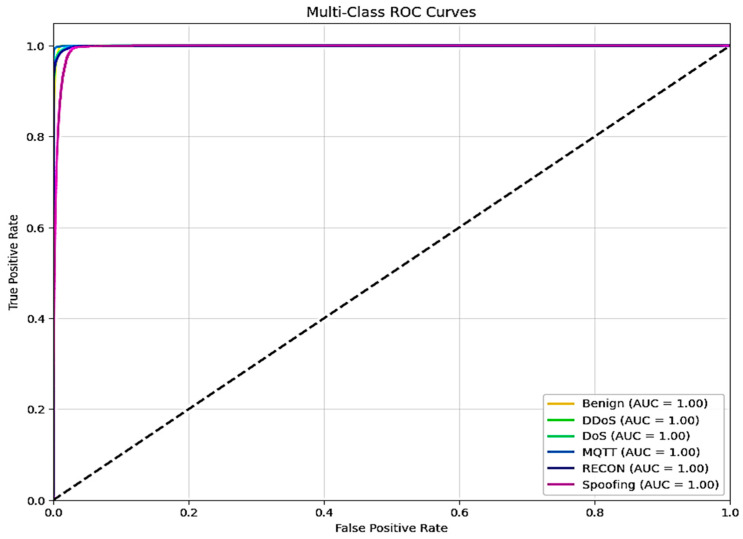
ROC curve for both proposed models for 6-class in CICIoMT 2024.

**Figure 28 sensors-25-03693-f028:**
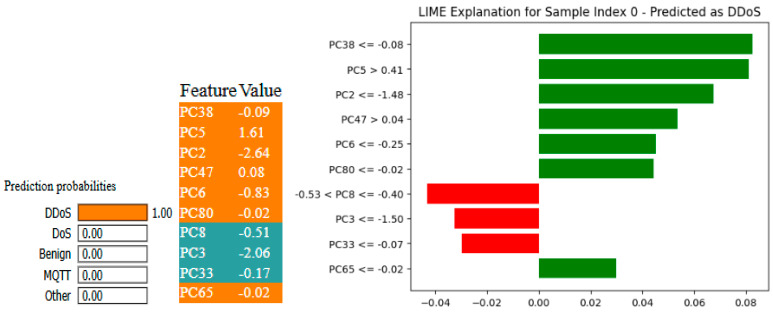
XAI-LIME explanations for XFuseRLSTM for 6-class in CICIoMT 2024.

**Figure 29 sensors-25-03693-f029:**
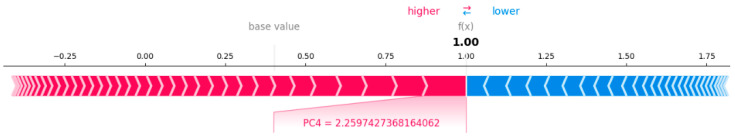
XAI-SHAP explanations for XFuseRLSTM for 6-class in CICIoMT 2024.

**Table 1 sensors-25-03693-t001:** Attack type mapping for NSL-KDD dataset.

Attack Category	Attack Type Mapped
Normal	Normal
DoS Attacks	apache2, back, land, mailbomb, Neptune, pod, processtable, smurf, teardrop, udpstorm
User to Root (U2R)	buffer_overflow, loadmodule, perl, rootkit, ps, xterm, sqlattack
Remote to Local (R2L)	ftp_write, guess_passwd, httptunnel, imap, multihop, named, phf, sendmail, snmpgetattack, snmpguess, spy, warezclient, warezmaster, xsnoop, xlock
Probe	ipsweep, mscan, nmap, portsweep, saint, satan

**Table 2 sensors-25-03693-t002:** Attack type mapping for CICIoTM 2024 dataset for 6-class.

Attack Category	Attacks Type Mapped
BENIGN	Benign_train
DDos	TCP_IP-DDoS- * which includes all 24 ICP_IP_Ddos attacks.
DOS	TCP_IP-DoS- * which includes all 16 ICP_IP_Dos attacks.
MQTT	It maps all 5 MQTT attacks, including DOS and Ddos
RECON	It maps all 4 types of Recon attacks
Spoofing	It maps to the ARP_Spoofing_train.

**Table 3 sensors-25-03693-t003:** Attack type mapping for the CICIoTM 2024 dataset for 19 classes.

Attack Category	Attacks Type Mapped
Benign	Benign_train
Spoofing	Same
MQTT-DoS-Connect_Flood	Same
MQTT-DoS-Publish_Flood	Same
MQTT-DDoS-Publish_Flood	Same
MQTT-DDoS-Connect_Flood	Same
MQTT-Malformed_Data	Same
Recon-OS_Scan	Same
Recon-Ping_Sweep	Same
Recon-Port_Scan	Same
Recon-VulScan	Same
DDoS-ICMP	TCP_IP-DDoS-ICMP * trains
DDoS-SYN	TCP_IP-DDoS-SYN * trains
DDoS-TCP	TCP_IP-DDoS-TCP * trains
DDoS-UDP	TCP_IP-DDoS-UDP * trains
DoS-ICMP	TCP_IP-DoS-ICMP * trains
DoS-SYN	TCP_IP-DoS-SYN * trains
DoS-TCP	TCP_IP-DoS-TCP * trains
DoS-UDP	TCP_IP-DoS-UDP * trains

**Table 4 sensors-25-03693-t004:** Comparison of the proposed models along with different domains.

Dataset	Model	Accuracy	Recall	Precision	F1 score	Loss	Training Time (Mins)	Epochs (Early Stopping)
Network	X-FuseDNN	94.34%	94.37%	94.72%	94.25%	0.1379	18	200 (50)
	X-FuseRLSTM	99.40%	99.40%	99.40%	99.40%	0.0266	3.1	200 (186)
NSL-KDD	X-FuseDNN	99.54%	99.54%	99.54%	99.54%	0.0199	5.67	200 (85)
	X-FuseRLSTM	99.72%	99.72%	99.72%	99.72%	0.0174	3.7	300 (222)
CICIoMT 2024 (19-class)	X-FuseDNN	97.27%	97.27%	97.27%	97.01%	0.0781	66.92	200 (55)
	X-FuseRLSTM	97.66%	97.66%	97.55%	97.46%	0.0796	100.1	200 (78)
CICIoMT 2024 (6-class)	X-FuseDNN	97.60%	97.60%	97.59%	97.59%	0.0636	57	200 (74)
	X-FuseRLSTM	98.05%	98.05%	98.02%	98.02%	0.0696	88.87	200 (86)

**Table 5 sensors-25-03693-t005:** Ablation study.

Dataset	Without PCA (Feature Fusion + RLSTM)	With PCA (Feature Fusion + LSTM)	With PCA and Skip Connections (X-Fuse RLSTM)
TON_IoT	99.23%	99.33%	99.40%
KDD	99.53%	99.67%	99.72%
CICIoMT (19-class)	97.09%	97.27%	97.66%
CICIoMT (6-class)	97.64%	97.87%	98.05%

**Table 6 sensors-25-03693-t006:** Comparison of X-FuseRLSTM with the benchmarked methods.

Dataset	References	Method Used	Accuracy
Network	Cao et al. [9]	Hybrid approach ML and DL	98.87%
	Wang et al. [10]	BHHO with CNN	99.10%
	Logeswari et al. [11]	Optimization with ML	93.70%
	Torre et al. [18]	CNN	98.20%
	Alabbadi and Bajaber [22]	CNN	99.24%
	Proposed X-FuseRLSTM	Feature Fusion+ Residual LSTM	99.40%
NSL-KDD	Dash et al. [12]	PSO-LSTMIDS, JAYA-LSTMIDS, and SSA-LSTMIDS	97.98%
	Xue et al. [13]	Hybrid LSTM + CNN	95.7%
	Raghunath et al. [14]	PSO + PCA + SVM	98.5%
	Imrana et al. [15]	CNN-GRU-FF	99.68%
	Thakkar et al. [16]	PCA + Autoencoders	82.22%
	Proposed X-FuseRLSTM	Feature Fusion+ Residual LSTM	99.72%
CICIMoT 2024	Akar et al. [17]	LSTM	98%
	Torre et al. [18]	CNN	97.31%
	Bo et al. [19]	Adaptive feature fusion	97.78%
	Proposed X-FuseRLSTM	Feature Fusion+ Residual LSTM	97.66% (19-class) 98.05% (6-class)

## Data Availability

The original contributions presented in this study are included in the article. Further inquiries can be directed to the corresponding author.

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
