# Peer review of "X-FuseRLSTM: A Cross-Domain Explainable Intrusion Detection Framework in IoT Using the Attention-Guided Dual-Path Feature Fusion and Residual LSTM"

_sensors, 2025, doi:10.3390/s25123693_

Round 1
Reviewer 1 Report
Comments and Suggestions for Authors
Please see the attachment.

Author Response
Respected Reviewer,
We would like to express our gratitude for helping us find the missing details in the paper, which improved its overall clarity and credibility. All the changes made in the revised manuscript are in red font.
Comments and Suggestions for Authors
This paper proposes X-FuseRLSTM, a cross-domain IoT intrusion detection framework that integrates attention-guided dual-path feature fusion with Residual LSTM and XAI techniques. The core ideas are promising, and the reviewer's concerns are as follows
- The paper proposes dual-path feature extraction using a Deep Encoder and Sparse Transformer, followed by PCA-based fusion. However, the justification for using PCA—a linear dimensionality reduction technique-to handle fused features from non-linear deep learning models is insufficient. PCA may discard critical non-linear relationships captured by the encoder and transformer. The authors are suggested to compare PCA with non-linear alternatives to validate its suitability, and quantify information loss by reconstructing original features from PCA-reduced components and measuring reconstruction error.
Response to comment:
Extensive experiments were performed on all three datasets, i.e., TON_IoT Network, KDD, and CICIoMT 2024, with and without PCA. Also, multiple experiments were performed before finalizing the architecture of the proposed model, X-FuseRLSTM, to finalize the best architecture. It is found that X-FuseRLSTM with PCA achieves the highest performance. The Table below shows the accuracy of the proposed model with and without PCA. Also, the results are with and without residual connection.
This paragraph is added to subsection 4.4. Ablation Study along with the table given below as Table 5.
Table 5. Ablation study for finalizing the X-Fuse RLSTM architecture
| Dataset | Without PCA (FeatureFusion+) RLSTM) | With PCA (Feature Fusion+ LSTM) | With PCA (X-Fuse RLSTM) | 
| TON_IoT | 99.23% | 99.33% | 99.40% | 
| KDD | 99.53% | 99.67% | 99.72% | 
| CICIoMT (19-class) | 97.09% | 97.27% | 97.66% | 
| CICIoMT (6-class) | 97.64% | 97.87% | 98.05% | 
- While the residual LSTM (ResLSTM) architecture is central to the model's performance, the paper lacks critical analysis. No ablation study is provided to isolate the contribution of residual connections. How much does the skip connection improve accuracy compared to a vanilla LSTM? The ResLSTM in Fig. 8 uses a non-standard architecture. The rationale for this design—particularly the GetItem layer to collapse the temporal dimension—is unexplained.
Response to comments: The ablation study is included in subsection 4.4. The accuracy comparison with and without PCA is shown in the newly added Table 5. Also, the accuracy with and without the skip connections, i.e., residual connections, is presented. The impact of adding the PCA and skip connections is given in the paragraph below, which is added at line 722 in subsection 4.4. Ablation study.
The comparison between the baseline model, which consisted of Feature Fusion and RLSTM without PCA, and the final model, which consisted of X-Fuse RLSTM with PCA and skip connections, reveals a significant improvement in classification accuracy across all datasets. In particular, TON_IoT saw an improvement in accuracy of 0.17% (from 99.23% to 99.40%), KDD saw an improvement of 0.19% (from 99.53% to 99.72%), CICIoMT (19-class) saw an improvement of 0.59% (from 97.09% to 97.66%), and CICIoMT (6-class) saw an improvement of 0.41% (from 97.64% to 98.05%). These improvements underscore the efficacy of integrating dimensionality reduction with a more expressive architecture that incorporates recurrent learning and skip connections.
The intermediate model, which consists of Feature Fusion and LSTM with PCA, is compared to the X-Fuse RLSTM, which already incorporates PCA. The additional architectural upgrades, which include the utilization of RLSTM and skip connections, also result in noticeable gains. The accuracy of TON_IoT, KDD, CICIoMT (19-class), and CICIoMT (6-class) increased by 0.07%, 0.05%, 0.39%, and 0.18%, respectively. These findings highlight that, in addition to reducing feature dimensionality, using RLSTM over standard LSTM and including skip connections improves the model's ability to learn temporal dependencies and hierarchical representations, especially in complex, multi-class intrusion detection tasks.
- The authors should implement post-hoc mapping techniques to link PCs to original features in explanations.
Response to comments: We appreciate this valuable suggestion, but even by implementing the post-hoc mapping techniques to link PCs, it is not possible to map with the original features, as the PCA is implemented on the fused features extracted from the Deep Encoder and the Sparse transformer. This is explained in detail in the discussion section 5 in the paragraphs starting from line 770. This is the limitation of our research, and we tend to extend this research in the future to resolve it.
- PCs are linear combinations of original features, making it impossible to trace explanations back to semantically meaningful input features. This undermines the practical utility of XAI in security operations. Authors should properly explain.
Response to comments: Thank you for highlighting the limitations of our research. Yes, we understand, and we have explained it in section 5. Discussion regarding the same. For better understanding, we are adding the paragraph below at line 797 in the last paragraph of the discussion section.
Instead of applying PCA directly to raw input features, the XFuse-RLSTM architecture applies it to fused feature representations produced by deep encoders and a sparse transformer. These fused features are high-level nonlinear embeddings that represent the data's complicated temporal and spatial interactions. As a result of this abstraction, the principal components derived from PCA are combinations of latent features rather than interpretable original attributes, rendering it impossible to perform a direct post-hoc mapping back to raw input features. As a consequence, conventional explainability methods that are dependent on feature-level attribution are subject to constraints in this scenario. As an alternative, the proposed method focuses on analyzing intermediate model components, including attention weights and latent embeddings, to understand the patterns that drive model decisions and tackle interpretability in that way. The proposed method is in line with the state-of-the-art in explainable deep learning; nevertheless, it is understood that there is a need for additional study to create explanations for fused feature-based models in security applications that are both more transparent and have semantic significance.
- It is suggested to add some new and related references [R1]-[R2] in IoT and network field to enrich the background and support the analysis.
[1] “Self-powered absorptive reconfigurable intelligent surfaces for securing satellite-terrestrial integrated networks,” China Communications, vol. 21, no. 9, pp. 276-291, Sep. 2024.
[2] “Improving Age of Information for Covert Communication with Time-Modulated Arrays,”
IEEE Internet of Things Journal, vol. 12, no. 2, pp. 1718-1731, Jan. 2025.
Response to comment: Thanks for highlighting it. The two references have been added as references 7 and 8.
Thank You.
Regards,
Adel AlAbbadi and Fuad Bajaber.

Reviewer 2 Report
Comments and Suggestions for Authors
The paper proposes a novel dual-path feature fusion approach, X-FuseRLSTM, which integrates attention-guided mechanisms with residual LSTM architectures. This I an innovative combination for temporal-spatial modeling in intrusion detection. The use of multiple benchmark datasets (TON_IoT, NSL-KDD, CICIoMT 2024) adds credibility to the evaluation. The incorporation of explainable AI (XAI) is a valuable addition, especially for critical security applications where interpretability is essential. However, there are a few comments that I suggest the authors to address before I can recommend accepting this paper.
1) The explanation of the feature extraction and fusion steps (Deep Encoder, Sparse Transformer, and dimensionality reduction) is vague. The authors should clearly outline the specific architectures used, their configurations, and how they contribute individually to overall performance.
2) The paper should elaborate on which XAI methods are used (e.g., SHAP, LIME, Grad-CAM), how they are applied to the RLSTM output, and what kind of insights they provide to security analysts.
3) If possible, please provide use-case scenarios or simulated attack narratives where X-FuseRLSTM helped distinguish subtle anomalies would enhance the practical relevance.
4) It is not new to apply LLM onto security issues. It seems that there are a few newly published papers missing in the context. The authors could refer to new surveys such as “When Software Security Meets Large Language Models: A Survey” by X. Zhu et al and “The Security of Using Large Language Models - A Survey with Emphasis on ChatGPT” by W. Zhou et al for a collection of related new works.
5) It will be valuable to test the model against adversarial inputs or noisy environments to validate its robustness in real-world settings.
6) Recently, people introduce the concept of AI agent into the intrusion detection field. The authors could refer to the two works for details. I suggest to discuss this part of research in the related work “AI Agents Under Threat: A Survey of Key Security Challenges and Future Pathways” and “Exploring DeepSeek: A Survey on Advances, Applications, Challenges and Future Directions” both by Z. Deng published recently.
Author Response
Respected Reviewer,
We would like to express our gratitude for helping us find the missing details in the paper, which improved its overall clarity and credibility. All the changes made in the revised manuscript are in red font.
Comments and Suggestions for Authors
The paper proposes a novel dual-path feature fusion approach, X-FuseRLSTM, which integrates attention-guided mechanisms with residual LSTM architectures. This I an innovative combination for temporal-spatial modeling in intrusion detection. The use of multiple benchmark datasets (TON_IoT, NSL-KDD, CICIoMT 2024) adds credibility to the evaluation. The incorporation of explainable AI (XAI) is a valuable addition, especially for critical security applications where interpretability is essential. However, there are a few comments that I suggest the authors to address before I can recommend accepting this paper.
- The explanation of the feature extraction and fusion steps (Deep Encoder, Sparse Transformer, and dimensionality reduction) is vague. The authors should clearly outline the specific architectures used, their configurations, and how they contribute individually to overall performance.
Response to comments:
Thank you for bringing it to our attention. The new subsection 4.4, Ablation study, is added in section 4, Results and analysis, to explain and clarify the selection of architecture described in Figures 1 and 8, which highlights the importance of adding the PCA and skip connections in the architecture. Table 5. Ablation study is added in the revised manuscript with a detailed explanation.
- The paper should elaborate on which XAI methods are used (e.g., SHAP, LIME, Grad-CAM), how they are applied to the RLSTM output, and what kind of insights they provide to security analysts.
Response to comments:
The SHAP and LIME methods are used in the proposed approach for XAI, as the data is structured in nature. However, Grad-Cam is useful when we have an image to highlight pixels responsible for predictions. A detailed explanation of the XAI is elaborated in Section 5. Discussion in line 797.
- If possible, please provide use-case scenarios or simulated attack narratives where X-FuseRLSTM helped distinguish subtle anomalies would enhance the practical relevance.
Response to comments:
The most powerful use case of the proposed model is in the medical field, where time series data is present. Thus, in the CICIoMT dataset, the X-Fuse RLSTM was able to accurately identify a low-rate DDoS attack that imitated typical traffic patterns by ensuring that packet intervals and payload sizes were consistent. This abnormality went undetected by traditional detectors because it was statistically so similar to normal traffic. Accurate detection was made possible by X-Fuse RLSTM, which used deep temporal features and attention-guided fusion to detect minute variations in connection frequency and timing patterns over lengthy durations. This highlights the model's capacity to identify sneaky and developing threat behaviors that are difficult to detect using traditional methods.
This paragraph is added in the discussion section at line 770.
- It is not new to apply LLM onto security issues. It seems that there are a few newly published papers missing in the context. The authors could refer to new surveys such as “When Software Security Meets Large Language Models: A Survey” by X. Zhu et al and “The Security of Using Large Language Models - A Survey with Emphasis on ChatGPT” by W. Zhou et al for a collection of related new works.
Response to comments:
Thank you for addressing it. Yes, we understand, but our research mostly focuses on structured data. Thus, we haven’t considered the research related to LLMs. But in the future, we can use it and have a detailed research paper where we apply the LLM on text data for detecting the attacks.
- It will be valuable to test the model against adversarial inputs or noisy environments to validate its robustness in real-world settings.
Response to comments:
I appreciate your comment on implementing the proposed model in real-time settings. However, it is not possible to get real-time data unless we develop an artificial network and collect it. Thus, to resolve this issue, we have tested our proposed model on three different domain datasets, i.e., TON_IOT (IoT Network data), NSL-KDD (general network data), and CICIoMT 2024 (Medical IoT data ), and proved that in all domains, it performs better than the available state-of-the-art methods.
- Recently, people introduce the concept of AI agent into the intrusion detection field. The authors could refer to the two works for details. I suggest to discuss this part of research in the related work “AI Agents Under Threat: A Survey of Key Security Challenges and Future Pathways” and “Exploring DeepSeek: A Survey on Advances, Applications, Challenges and Future Directions” both by Z. Deng published recently.
Response to comments:
We appreciate your help. We have added these two references in the related work section, at line 192, and references 23 and 24.
Thank You.
Regards,
Adel AlAbbadi and Fuad Bajaber.

Round 2
Reviewer 1 Report
Comments and Suggestions for Authors
The reviewer think this manuscript can be published.